# The Effect of Trabecular Aspiration on Intraocular Pressure, Medication and the Need for Further Glaucoma Surgery in Eyes with Pseudoexfoliation Glaucoma

**DOI:** 10.3390/diseases12050092

**Published:** 2024-05-06

**Authors:** Verena Prokosch, Sarah B. Zwingelberg, Desislava V. Efremova, Francesco Buonfiglio, Norbert Pfeiffer, Adrian Gericke

**Affiliations:** 1Department of Ophthalmology, University of Cologne, Kerpener Str. 62, 50937 Köln, Germany; verena.prokosch@uk-koeln.de (V.P.); sarah.zwingelberg@uk-koeln.de (S.B.Z.); 2Department of Ophthalmology, University Medical Center, Johannes Gutenberg University Mainz, Langenbeckstr. 1, 55131 Mainz, Germany; desi_ve@yahoo.com (D.V.E.); fbuonfig@uni-mainz.de (F.B.); norbert.pfeiffer@unimedizin-mainz.de (N.P.)

**Keywords:** pseudoexfoliation, glaucoma, intraocular pressure, trabecular aspiration, trabeculectomy

## Abstract

Purpose: To investigate whether trabecular aspiration (TA) has an effective medium-term intraocular pressure (IOP)-lowering and medication-saving effect in patients with pseudoexfoliation glaucoma (PEG). In addition, a subgroup analysis of patients with or without a previous trabeculectomy was performed. Methods: Records of 290 consecutive eyes with PEG that underwent TA between 2006 and 2012 at the Department of Ophthalmology, Mainz, Germany, were retrospectively analyzed with a follow-up period of 3 years. The main outcomes were IOP and the need for further medical treatment. Results: Of the 290 eyes with PEG that received TA, 167 eyes from 127 patients met the inclusion criteria. Among these eyes, 128 received TA and cataract surgery (Phaco-TA) without having had a trabeculectomy (group I) before, 29 had Phaco-TA after a previous trabeculectomy (group II) and 10 underwent stand-alone TA after a previous trabeculectomy (group III). In the whole cohort, the median IOP decreased immediately after TA and remained significantly lower compared to the baseline throughout the period of 36 months. Likewise, the median number of antiglaucoma drugs was reduced over the whole period. At the same time, in group I, the median IOP and the number of antiglaucoma drugs were reduced over 36 months. In contrast, in the post-trabeculectomy groups (group II and III), the median IOP and the number of antiglaucoma drugs could not be reduced. While most of the patients that received Phaco-TA with or without a previous trabeculectomy (group I and II) did not require further surgical intervention during the follow-up period, almost all patients receiving stand-alone TA after a previous trabeculectomy (group III) needed surgical therapy, most of them between the second and the third year following TA. Conclusions: Phaco-TA has an effective medium-term pressure-lowering and medication-saving effect, especially in patients without a previous trabeculectomy. In trabeculectomized eyes, the effect of TA is limited but still large enough to delay more invasive surgical interventions in some patients.

## 1. Introduction

The global prevalence of glaucoma was estimated to be 76 million in 2020 and is projected to increase to 112 million by 2040 [1]. Pseudoexfoliation glaucoma (PEG) is the most common variant of secondary open-angle glaucoma [2,3]. Particularly in Northern Europe, the prevalence of PEG is reported to be significantly elevated [2,3,4,5,6,7,8,9]. In patients with PEG, the risk of blindness is increased due to secondary optic nerve atrophy caused by intraocular pressure (IOP) elevation, with severe pressure spikes due to amyloidosis-like protein deposition in the anterior segment of the eye [10,11]. Importantly, PEG represents the ocular manifestation of pseudoexfoliation syndrome, an age-related degenerative systemic disorder [12]. This syndrome stems from the generation and accumulation of aberrant fibrillar material, which deposits in various human organs, including the heart, kidneys, lungs, gallbladder, liver and meninges. Consequently, it is associated with the onset of notable pathological conditions such as aortic aneurysm, hypertension, dementia and cerebral atrophy [13,14,15].

IOP is a crucial factor associated with the risk and progression of glaucoma and lowering IOP often delays disease progression [16,17,18,19]. Various approaches exist to reduce IOP. In PEG, trabecular aspiration (TA), a less invasive surgery aimed at removing the pseudoexfoliation material in the chamber angle to facilitate aqueous humor outflow, is one proposed method [11,20]. For TA, a specially designed cannula is used to aspirate the pseudoexfoliation material with suction of 100 to 200 mmHg in patients with PEG. This procedure is typically combined with cataract surgery to avoid a two-stage surgery, reducing the risk of complications and irritation. However, the feasibility of this technique in patients after a trabeculectomy is unknown. It is rather believed that the trabecular meshwork does not filter much aqueous humor once shut down after a trabeculectomy [21,22].

The aim of the present study was to investigate the medium-term effectiveness of TA in lowering IOP and reducing the need for medication in patients with PEG. Additionally, a subgroup analysis was performed, considering some patients who had already undergone a trabeculectomy and others who received TA combined with phacoemulsification.

## 2. Materials and Methods

### 2.1. Data Collection

In this retrospective observational study, the medical records of 290 patients treated for PEG between 2006 and 2012 at the Department of Ophthalmology, University Medical Center of the Johannes Gutenberg University Mainz, Germany, were reviewed for TA procedures. The minimum follow-up period after surgery was 6 months. Ethical approval was not required for this retrospective analysis, as per local law (“Landeskrankenhausgesetz” §36, §37). The indication for TA was to reduce IOP in patients with intraocular pseudoexfoliation material exceeding their individual target pressure or in the case of progression of glaucoma damage. The progression of glaucoma damage was determined based on morphological or functional criteria. Morphological changes were determined by the evaluation of consecutive optic disc photographs, by confocal scanning laser ophthalmoscopy or scanning laser polarimetry or by optical coherence tomography.

Progress in visual field loss was defined as a reproducible loss in mean deviation observed across three successive follow-up appointments. An overall loss exceeding 1 dB, a reduction of 10 dB in a single measured point or declines of more than 5 dB in three points were deemed significant. Another indication for TA was to reduce the number of antiglaucoma drugs in patients with either an intolerance to specific agents or because of insufficient adherence. Since 2012, TA has only been performed infrequently and/or in a modified fashion in our clinic due to the introduction of a variety of other microinvasive glaucoma surgery (MIGS) procedures affecting aqueous humor outflow. Therefore, we did not include patients receiving TA after 2012.

### 2.2. Inclusion and Exclusion Criteria

The inclusion criteria were as follows: patients with a diagnosis of PEG, patients who underwent TA alone before and after a trabeculectomy and patients who underwent combined TA and cataract surgery, either with or without a previous trabeculectomy. Documentation of IOP, measured by Goldmann applanation tonometry, preoperatively and at least 36 months postoperatively, through two daily tension profiles, was required, as well as documentation of medication use from preoperatively to at least 36 months postoperatively.

Patients were excluded from the study if there was insufficient postoperative documentation after patient discharge and insufficient postoperative follow-up of at least 36 months. Failure to receive a response or documentation from the referring physician within 60 days of contact was also considered an exclusion criterion.

For the subgroup analysis, patients were divided into three groups based on their surgical history:Group I—Eyes receiving combined TA plus cataract surgery (Phaco-TA) without a previous trabeculectomy;Group II—Eyes receiving Phaco-TA after a previous trabeculectomy;Group III—Eyes receiving TA as a primary surgical intervention after a trabeculectomy.

Notably, there was no patient who received TA as a single procedure, without having had a previous trabeculectomy.

### 2.3. Clinical Information and Outcome Parameters

Demographic data, including age and gender, were recorded for each patient. IOP, drug therapy and details of the surgical interventions were noted at various time points: 30 days and 1 day preoperatively, and on the 1st, 2nd, 3rd, 7th, 14th and 30th days and at 3, 6, 12, 18, 24, 30 and 36 months postoperatively. The primary outcome measure was IOP following TA. Secondary outcome measures were the number of antiglaucoma agents used and the need for further surgical interventions after TA.

### 2.4. Survival Analysis

To determine the stability of glaucoma after TA, a Kaplan–Meier analysis was performed. If, following TA, a surgical IOP-lowering intervention was required due to exceeding the target IOP or the morphological and/or functional progression of glaucoma, the treatment by TA was regarded as failed.

### 2.5. Statistical Analysis

The data analysis was performed using GraphPad Prism 6.0 (GraphPad Inc., San Diego, CA, USA). The Shapiro–Wilk normality test was conducted to test for the data distribution. Since the data were not normally distributed, they are presented as boxplots with minimum and maximum values. The Wilcoxon test was used to compare IOP and medication scores over time with preoperative values. Missing values were handled using the Last-Observation-Carried-Forward (LOCF) method. Since 14 comparisons were made with the preoperative value at 30 days prior to surgery, the level of significance was reduced to 0.05/14 = 0.0036 by using the Bonferroni correction. Descriptive presentations were based on the available complete case analysis, and there were no significant differences when comparing the existing data to the LOCF data in the statistical tests. The survival probability was calculated by Kaplan–Meier survival analysis. For comparisons of the Kaplan–Meier curves, the log-rank (Mantel-Cox) test and the Gehan–Breslow–Wilcoxon test were used.

## 3. Results

### 3.1. Demographics

The study included 290 consecutive eyes with PEG. According to the inclusion and exclusion criteria, a total of 167 eyes from 127 patients that received TA were included in the study. After the analysis of the whole cohort, the eyes were distributed into three groups: group I (Phaco-TA without previous trabeculectomy, *n* = 128), group II (Phaco-TA after previous trabeculectomy, *n* = 29) and group III (TA stand-alone after trabeculectomy, *n* = 10). Of note, there was no eye that received TA as a primary procedure without having undergone a trabeculectomy before. The demographic data for each group are presented in Table 1.

### 3.2. Effect of TA on IOP

In the whole cohort, the initial median IOP was 18.0 (7.00–50.0) mmHg 30 days preoperatively. One day after surgery, IOP markedly decreased to 16.7 (5.00–38.7) mmHg and reached its lowest level of 14.0 (5.00–32.0) mmHg on the 3rd day after surgery. Thereafter, IOP level remained low throughout the period of 36 months to reach a median level of 15.0 (7.00–34.0) mmHg at 36 months after surgery (**** *p* < 0.0001; 36 months postoperatively versus 30 days preoperatively, Figure 1A).

To test whether previous or combined surgery affected IOP’s development after TA, we divided the data into three groups. The baseline IOP values differed slightly between the groups. While the median IOP was 18.0 (10.7–50.0) mmHg in group I (Phaco-TA without previous trabeculectomy), it was 16.0 (7.00–25.4) mmHg in group II (Phaco-TA after previous trabeculectomy) and 22.5 (12.0–32.0) mmHg in group III (stand-alone TA after trabeculectomy; ** *p* < 0.01; group III versus group II, Kruskal–Wallis and Dunn’s multiple comparisons post hoc test).

In group I, the time course of the median IOP was similar to that of the whole cohort. The median IOP 30 days prior to surgery was 18.0 (10.7–50.0) mmHg and markedly decreased to 17.0 (5.00–38.7) mmHg already on the first day after surgery to reach its minimum of 14.0 (5.00–32.0) mmHg on the 3rd day after surgery. Afterwards, the IOP level remained low and was 15.0 (8.00–34.0) mmHg 36 months after surgery (**** *p* < 0.0001; 30 days preoperatively versus 36 months postoperatively, Figure 1B).

Remarkably, Phaco-TA after trabeculectomy had no significant effect on the median IOP values. While the median IOP 30 days preoperatively was 16.0 (7.00–25.4) mmHg, it only decreased by tendency to 13.0 (6.00–26.0) mmHg on the first day after surgery and to 15.0 (6.00–25.0) mmHg on the 3rd day after surgery. After 36 months, the median postoperative IOP was 15.0 (7.00–30.0) mmHg (*p* > 0.05; Figure 1C). Likewise, the median IOP in trabeculectomized eyes that received stand-alone TA did not decrease significantly. While, in this group, the initial IOP 30 days prior to surgery was 22.5 (12.0–32.0) mmHg, it only decreased by tendency to 16.0 (7.00–30.0) mmHg on the first postoperative day and to 17.5 (8.00–26.0) mmHg on the 3rd postoperative day (*p* > 0.05, Figure 1D). At the end of the follow-up time, the median IOP was 22.0 (11.0–34.0) mmHg in this group.

### 3.3. Effect of TA on the Number of Antiglaucoma Agents Used

The patients applied antiglaucoma medications spanning five distinct categories: β-adrenoceptor blockers, α_2_-adrenoceptor agonists, prostaglandin analogs, carbonic anhydrase inhibitors and parasympathomimetics.

In the whole cohort, the initial median number of antiglaucoma drugs was 1.00 (0.00–4.00) 30 days preoperatively. One and 3 days after surgery, the number decreased to 0.00 (0.00–3.00), respectively, and was 1.00 (0.00–4.00) 36 months postoperatively (**** *p* < 0.0001; 36 months postoperatively versus 30 days preoperatively, Figure 2A). When comparing the number of antiglaucoma agents 30 days preoperatively, there were no differences between the groups (*p* > 0.05; Kruskal–Wallis test).

In the group receiving Phaco-TA without previously having had a TE (group I), the median drug number was 2.00 (0.00–4.00) 30 days prior to surgery and markedly decreased to 0.00 (0.00–3.00) on the first and third postsurgical day, respectively. After 36 months, the median number of antiglaucoma medications was 1.00 (0.00–3.00) (**** *p* < 0.0001; 36 months postoperatively versus 30 days preoperatively, Figure 2B). 

In the group receiving Phaco-TA after having received a trabeculectomy (group II), the initial number of antiglaucoma drugs was 1.00 (0.00–3.00) 30 days preoperatively and markedly decreased to 0.00 (0.00–2.00) on the 1st and 3rd day after surgery. Remarkably, the number of medications was significantly lower until the 1st month after surgery and then reached the initial levels (Figure 2C). 

In eyes that received stand-alone TA after previously having received a TE (group III), the initial number of antiglaucoma agents at 30 days prior to surgery was 2.00 (0.00–4.00) and postoperatively only reduced by tendency (Figure 2D).

### 3.4. Effect of TA on Further Surgical Interventions

The Kaplan–Meier analysis revealed that, of all 167 treated eyes, 82% did not require further surgical intervention during the follow-up period of 36 months (Figure 3A). 

The subgroup analysis revealed that 88% of the eyes that received Phaco-TA without a previous trabeculectomy did not require further surgical intervention during the whole follow-up period of 36 months (Figure 3B). Of the eyes that received Phaco-TA after a previous trabeculectomy 79% did not require further IOP-lowering surgery during the follow-up period. The survival rate did not differ significantly from that of the group that received Phaco-TA without a trabeculectomy (Figure 3B). Remarkably, in the group that received TA only after a trabeculectomy, only 1 of 10 eyes (10%) did not require further surgery during the follow-up period of 36 months (Figure 3B). However, it should be mentioned that, in this group, all patients received surgery because the target pressure was exceeded, and, 24 months after TA, 60% of the eyes still did not require further IOP-reducing surgery, suggesting that TA may be useful to delay the use of more invasive surgery in patients who are above their target pressure levels after a trabeculectomy.

## 4. Discussion

This study constitutes the inaugural assessment of the impact of TA over a 36-month period, with a specific focus on its enduring effects on IOP and a reduction in antiglaucoma medications. Additionally, a subgroup analysis was conducted to scrutinize patients who had previously undergone a trabeculectomy. This study has yielded several noteworthy findings. First, our investigation of 167 eyes affirms the effectiveness of TA as a therapeutic approach in terms of reducing IOP and minimizing the use of antiglaucoma medications over the course of 36 months. Second, in eyes that had previously undergone a trabeculectomy, whether through Phaco-TA or stand-alone TA, the impact on IOP reduction and medication usage was more modest. Nevertheless, in patients with a history of trabeculectomy, TA presents itself as a minimally invasive surgical alternative capable of postponing more invasive procedures.

The elevation of IOP in PEG can be attributed to the congestion of the trabecular meshwork due to the accumulation of pseudoexfoliation material and pigment granules. These accumulations induce morphological alterations in the trabecular lamellae [11], including the deposition of extracellular matrix components resulting from the release of growth factors such as basic fibroblastic growth factor, connective tissue growth factor and matrix metalloproteinases (including MMP-2, MMP-9 and MMP-14). These processes eventually lead to the fibrosis of the trabecular meshwork [23].

TA has been proposed as a causative therapy to address the removal of these deposits. According to Jacobi and Krieglstein, TA can be viewed as an effective and IOP-lowering procedure capable of eliminating intertrabecular and pretrabecular deposits within the trabecular meshwork, thereby proving efficacious in treating PEG [20,24]. Studies investigating the use of TA as a stand-alone treatment for PEG have reported favorable short- to mid-term reductions in IOP. For instance, Jacobi and Krieglstein observed a sustained decrease in IOP, lowering it from a preoperative mean of 37.4 mmHg to 18.3 mmHg 15 months after surgery [24]. Similarly, Grüb et al. reported a notable decrease in IOP, from 26.8 mmHg preoperatively to 19.1 mmHg one month after TA [25]. However, this effect was transient and typically lasted only a few weeks in most patients. In a comparison by Jacobi et al., the IOP-lowering effect of stand-alone TA was assessed alongside Phaco-TA in PEG patients. The results revealed a reduction in the mean IOP from 31.4 to 19.0 mmHg after 2 years with stand-alone TA and a reduction from 32.4 to 18.7 mmHg 2 years after Phaco-TA [26]. Notably, Phaco-TA exhibited superior effectiveness in terms of IOP reduction. Supporting this assertion, a study by Georgopoulos and colleagues demonstrated that Phaco-TA achieved better IOP control and necessitated fewer postoperative medications compared to cataract surgery alone [27]. As a result, Phaco-TA has solidified its position as a valuable tool in achieving a significant reduction in IOP among patients with PEG. Dinslage and colleagues, in their study, demonstrated a mean IOP reduction from 25.4 mmHg to 17.0 mmHg at the 2-year mark after Phaco-TA [28]. Similarly, Klamann and associates reported a decrease in the mean IOP from 22.2 mmHg to 17.1 mmHg one year after Phaco-TA [29]. Widder et al., in another investigation, documented a reduction in IOP from 25.3 mmHg to 18.1 mmHg at the 15-month post-intervention interval [30]. It is important to note that the follow-up duration in these studies ranged from 1 to 2.5 years.

Our present study extends these previously reported findings by revealing a consistent and highly significant reduction in the median IOP in a large cohort of patients, which decreased from 19.5 mmHg to 15.1 mmHg (with a range of 10.7–50.0) over the course of 36 months following TA. Moreover, our findings demonstrate that the IOP-lowering effect of TA is reduced in eyes that have previously received a trabeculectomy.

It is worth emphasizing that cataract surgery alone exerts a moderate IOP-lowering effect [31,32,33,34,35]. In fact, this surgical procedure significantly reduces IOP in PEG patients. Merkur et al. investigated the postoperative IOP-lowering effects of stand-alone cataract surgery in various patient groups (PEG, primary open-angle glaucoma and cataract), revealing a more substantial long-term IOP reduction in the PEG group compared to both the primary open-angle glaucoma group and the cataract-alone control group [36]. Consistent with this study, a recent investigation by Ramezani and colleagues, examining the IOP-lowering effects of cataract surgery in PEG and cataract patients, observed a greater postoperative IOP decrease in patients with both cataract and PEG (mean IOP difference pre- and post-operative of −4.5 mmHg) compared to patients with cataract alone and no glaucoma (mean IOP difference pre- and post-operative of −2.3 mmHg) [37]. More recently, Rao and Cruz conducted a comparative case series study involving PEG patients, revealing that stand-alone cataract surgery can be as effective as cataract surgery combined with trabeculectomy in terms of long-term IOP reduction [38].

Regarding the impact of surgical interventions on postoperative antiglaucoma therapy, Grüb and colleagues observed a decrease in therapeutic requirements following TA alone, with a reduction in medication usage from 3.1 ± 0.9 to 0.8 ± 1.2 at 180 days after the procedure [25]. In line with these findings, Jacobi and Krieglstein also reported a decrease, indicating a preoperative mean requirement for antiglaucoma medications before stand-alone TA of 4.3 ± 1.9, which was reduced to 1.4 ± 0.4 at the 15-month post-surgery assessment [24]. Furthermore, Jacobi et al. also investigated postoperative antiglaucoma therapy, revealing that, in the Phaco-TA group, 54% of patients no longer required medication control at the 2-year post-surgery mark, while, in the stand-alone TA group, 45% of patients had ceased medication use by 18 months following the intervention [26]. Moreover, Georgopoulos and colleagues, in their comparison of Phaco-TA and cataract surgery as a stand-alone procedure, demonstrated a decrease in the use of antiglaucoma medications from 1.5 ± 0.5 to 0.4 ± 0.6 by 12–18 months after Phaco-TA, whereas the reduction was from 1.4 ± 0.5 to 1.0 ± 0.8 at 12–18 months after cataract surgery alone [27]. Additionally, Klamann et al. conducted an assessment and found no statistically significant difference in the number of antiglaucoma medications required between Phaco-TA and cataract surgery combined with trabectome at any point after surgery [29]. After 15 months of follow-up, Widder and co-workers reported a reduced medication score, which decreased from 3.8 ± 1.8 to 2.3 ± 1.5 [30].

In our study, the Phaco-TA group exhibited a significant reduction in medication usage during the whole postoperative follow-up period of 36 months. Remarkably, in patients who received Phaco-TA after a previous trabeculectomy, the mean medication score also decreased significantly in the first postoperative month but subsequently increased to nearly preoperative levels within the observation period. In eyes that had previously undergone a trabeculectomy, stand-alone TA yielded negligible effects in terms of IOP reduction and in reducing the number of antiglaucoma drugs.

A plausible explanation for this phenomenon may be found in the anatomical characteristics of the eye and the postoperative scarring events that trigger substantial remodeling processes, drastically altering the structure of the trabecular meshwork (TM) and associated pathways [39,40]. It is conceivable that the diminished effectiveness of TA after a trabeculectomy has not been previously explored in studies, given that the outflow system through the TM, Schlemm’s canal and the canaliculi undergo degeneration, adhesion and obliteration when bypassed by trabeculectomy. Over time, following a trabeculectomy, it is possible that the outflow pathway through the TM becomes obstructed by pseudoexfoliation (PEX) material. Additionally, further fibrotic events may diminish the impact of TA after a trabeculectomy [41,42]. Our findings align with this hypothesis, suggesting that the outflow through the TM may be reactivated only for a short period in some patients.

Taken together, our results provide valuable insights into the effectiveness of Phaco-TA in reducing the intraocular pressure (IOP) and diminishing the need for hypotensive medications among pseudoexfoliation glaucoma (PEG) patients without a prior history of trabeculectomy.

Nevertheless, several limitations should be considered. First, our study relied on retrospective record analysis, which comes with inherent limitations. Furthermore, the three groups differed in terms of their baseline IOP levels. The stand-alone TA group consisted of patients with notably elevated IOP values, some of whom were already on the maximum medication or had contraindications for certain antiglaucoma agents. These patients were not inclined toward or suitable for more complex glaucoma surgery at the time of TA, which was the reason for choosing this low-invasive type of surgery. Interestingly, the survival analysis indicated that, in 60% of the cases, subsequent surgery could be deferred for at least 2 years, suggesting that TA had a stabilizing effect on IOP in these eyes. However, it is essential to note that we did not directly compare stand-alone cataract surgery patients and Phaco-TA patients, although it is well established that cataract surgery alone has a moderate IOP-lowering effect. However, for many years, Phaco-TA has been a standard treatment for PEG patients with simultaneous cataracts in our clinic, and no patient with PEG that received stand-alone cataract surgery could be identified. Additionally, we were unable to include patients who underwent stand-alone TA without a prior trabeculectomy in our analysis. Therefore, our ability to draw definitive conclusions regarding the exclusive effect of TA remains limited. However, we can draw the conclusion that stand-alone TA is suitable as a less invasive surgical intervention to postpone more invasive surgery in patients with PEG who have undergone a previous trabeculectomy, because most of the patients treated needed further IOP-reducing surgery between the second and third years of the follow-up period.

## 5. Conclusions

In summary, this study underscores the IOP-reducing and medication-sparing benefits of Phaco-TA in PEG, especially in patients without a history of trabeculectomy. As with any medical intervention, the decision to perform TA should be tailored to each patient’s unique circumstances and requirements. In selected cases, TA alone following a trabeculectomy may offer a practical means of postponing more invasive surgical interventions.

## Figures and Tables

**Figure 1 diseases-12-00092-f001:**
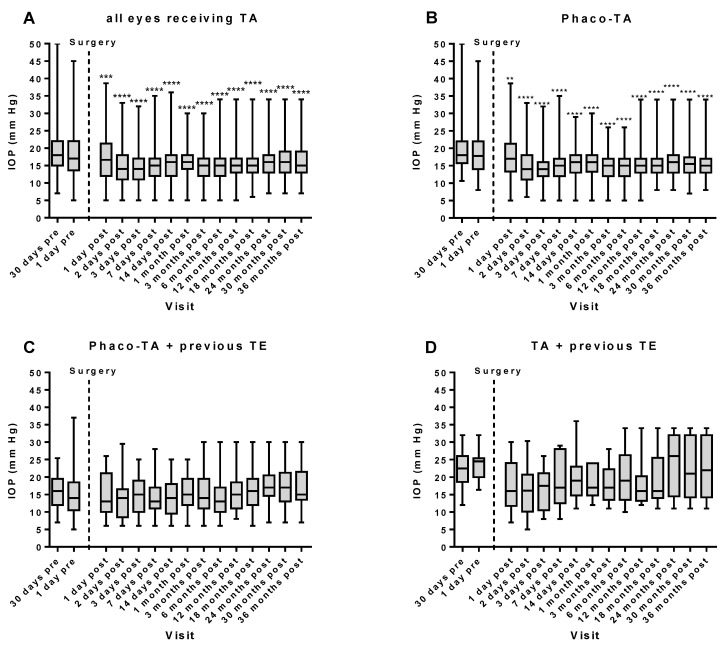
Time course of intraocular pressure (IOP) following trabecular aspiration (TA). In the whole cohort (*n* = 167), TA induced a significant and permanent reduction in IOP over the whole follow-up period of 36 months (**A**). In the group that received TA + cataract surgery (Phaco-TA, *n* = 128), a permanent reduction in IOP was also observed throughout the follow-up period (**B**). In contrast, eyes that received Phaco-TA after a previous trabeculectomy (TE, *n* = 29) did not display any substantial IOP reduction (**C**). Likewise, eyes that received stand-alone TA after a previous TE (*n* = 10) experienced only, by tendency, a transient reduction in IOP (**D**). ** *p* < 0.01; *** *p* < 0.001; **** *p* < 0.0001; IOP at a postoperative timepoint versus IOP 30 days preoperatively.

**Figure 2 diseases-12-00092-f002:**
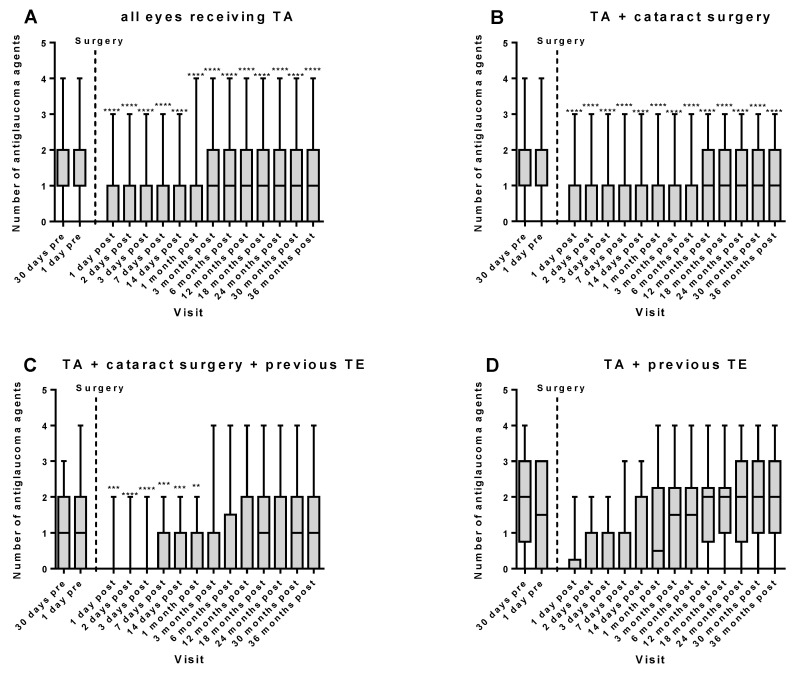
Number of antiglaucoma agents used before and after trabecular aspiration (TA) during follow-up period of 36 months. In the whole cohort (*n* = 167), a significant and permanent reduction in antiglaucoma drugs was achieved after TA (**A**). Likewise, in the Phaco-TA group (*n* = 128), a permanent reduction in drugs was achieved (**B**). In eyes that received Phaco-TA after a previous trabeculectomy (TE, *n* = 29), only a transient reduction in antiglaucoma drugs was achieved during the first month after surgery (**C**). Similar, a transient reduction by tendency in antiglaucoma medication was observed following stand-alone TA after a previous TE (*n* = 10, (**D**)). ** *p* < 0.01; *** *p* < 0.001; **** *p* < 0.0001; number of antiglaucoma agents at a postoperative timepoint versus number of antiglaucoma agents 30 days preoperatively.

**Figure 3 diseases-12-00092-f003:**
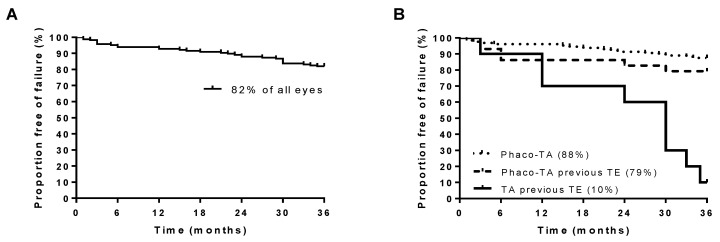
Kaplan–Meier analysis of eyes with regard to stability of glaucoma. (**A**) The survival analysis of all eyes treated with TA (*n* = 167) revealed that 82% of the eyes did not require surgical intervention to lower IOP within the follow-up period of 36 months. (**B**) Survival was, by tendency, slightly lower in eyes that received Phaco-TA with a previous trabeculectomy (TE) compared to those without a previous TE. At the end of the follow-up period, only 10% of the eyes that received TA after a previous TE did not require further surgery to reduce IOP.

**Table 1 diseases-12-00092-t001:** Overview of demographic data.

Group	Number of Eyes (*n*)	Age (Median, Range)	Female (%)	Male (%)
Whole cohort	167	79 (59–95 years)	68	32
Group I (Phaco-TA)	128	79 (59–94 years)	69	31
Group II (Phaco-TA + previous trabeculectomy)	29	78 (67–91 years)	69	31
Group III (TA + previous trabeculectomy)	10	80 (67–95 years)	68	32

## Data Availability

Data will be made available on request.

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
