# Peer review of "The Effect of Trabecular Aspiration on Intraocular Pressure, Medication and the Need for Further Glaucoma Surgery in Eyes with Pseudoexfoliation Glaucoma"

_diseases, 2024, doi:10.3390/diseases12050092_

Round 1

Reviewer 1 Report

Comments and Suggestions for Authors

The authors presented a very interesting study evaluating the effect of trabecular aspiration in  medium-term IOP-lowering and medication-saving in patients with pseudoexfoliation glaucoma. The manuscript is clear, its topic is original in content, and the conclusions are consistent with the evidence presented. The manuscript is with merit and the findings are worth reporting. Please address the following minor comments before publication:

o   How the IOP was measured? The authors should add this information in the methods

o   The authors should provide if available information about the functional (VF) and/or structural (OCT) severity degree of the disease in the study participants

o   How do the authors define “progressive visual field loss”?

o   Did the authors collect information regarding the type of antiglaucomatous drugs used (in addition to the number)?

Author Response

The authors presented a very interesting study evaluating the effect of trabecular aspiration in  medium-term IOP-lowering and medication-saving in patients with pseudoexfoliation glaucoma. The manuscript is clear, its topic is original in content, and the conclusions are consistent with the evidence presented. The manuscript is with merit and the findings are worth reporting. Please address the following minor comments before publication:

We thank the reviewer for the comments and the suggestions.

1.) How the IOP was measured? The authors should add this information in the methods

Response to 1.): Intraocular pressure was measured by Goldmann applanation tonometry. We added a statement accordingly (line 95, underlined).

2.) The authors should provide if available information about the functional (VF) and/or structural (OCT) severity degree of the disease in the study participants

Response to 2.): Our primary outcome measure was reduction of IOP by trabecular aspiration. Secondary outcome measures were the number of glaucoma agents and the need for further surgical interventions after trabecular aspiration. We included a respective statement in the Materials and Methods section (lines 112-114). Since the devices for assessing visual field (Octopus, Haag-Streit and Humphrey Field Analyzer, Zeiss) and structural changes (HRT, GDX, OCT) differed between individual patients, we do not provide this information. However, we included a section, describing the criteria used for determining glaucoma progression (lines 75-85).

3.) How do the authors define “progressive visual field loss”?

Response to 3.): Progress in visual field loss was defined as reproducible loss in mean deviation over three consecutive follow-up visits. An overall loss of more than 1dB, decrease in one measured point of 10 dB or 3 points of more than 5 dB was regarded as significant loss. We added a passage accordingly (lines 81-84, added section is underlined).

4.) Did the authors collect information regarding the type of antiglaucomatous drugs used (in addition to the number)?

Response to 4.): Yes, we collected the information. The drugs belonged to five distinct categories: β-adrenoceptor blockers, α2-adrenoceptor agonists, prostaglandin analogs, carbonic anhydrase inhibitors, and parasympathomimetics. We added a passage on this topic (lines 194-196).

Reviewer 2 Report

Comments and Suggestions for Authors

The authors studied the effect of trabecular aspiration in a special form of secondary open-angle
glaucoma called pseudoexfoliative syndrome, which can affect several other organs in addition to the
eye. A lot of work has gone into writing the manuscript.

It would be useful to add the non-ophthalmic aspects of pseudoexofialtive sy to the introduction

It is not clear from the manuscript:

what was the indication for trabecular aspiration? What was the indication for trabecular aspiration
especially in cases with low intraocular pressure? At which baseline intraocular pressure values were
achieved significant IOP reduction and visual field stabilization.? How did they measure of visual field
stabization?

The authors’ conclusion, namely trabecular aspiration has an effective medium-term pressure-
lowering and medication-saving effect especially in patients without previous trabeculectomy is not
supported. In the phaco combined trabecular aspiration group without trabeculectomy the lowering
effect could also be explained with the deepening of anterior chamber after the lens removal. Only if
a purely post-phaco pseudoxfoliative glaucomatous eye does not experience a significant reduction
in intraocular pressure could it be said that trabecular aspiration has an effect. Trabecular aspiration
might be a good adjuvant treatment possibility

Author Response

The authors studied the effect of trabecular aspiration in a special form of secondary open-angle glaucoma called pseudoexfoliative syndrome, which can affect several other organs in addition to the eye. A lot of work has gone into writing the manuscript.

We thank very much the reviewer for these comments.

1.) It would be useful to add the non-ophthalmic aspects of pseudoexofiative syndrome to the introduction

Response to 1.): We added a passage in the Introduction accordingly (lines 46-52, added passage is underlined).

2.) What was the indication for trabecular aspiration?

Response to 2.): The indication was to reduce intraocular pressure in patients with pseudoexfoliation exceeding their individual target pressure or in those with glaucoma progression. Another indication was to reduce the number of glaucoma drugs in patients with either intolerance to specific drugs or because of insufficient adherence. We added a statement accordingly (lines 75-89, added passage is underlined).

3.) What was the indication for trabecular aspiration especially in cases with low intraocular pressure?

Response to 3.): In cases with low IOP, the goal of trabecular aspiration was either to reduce intraocular pressure below the individual target pressure or to reduce the number of antiglaucomatous drugs. We added a statement accordingly (lines 75-89, added passage is underlined).

4.) At which baseline intraocular pressure values were achieved significant IOP reduction and visual field stabilization?

Response to 4.): There was not specific value at which significant IOP reduction and visual field stabilization was achieved. In many patients, we did the procedure even at low intraocular pressure levels to reduce the number of glaucoma drugs (see lines 75-89).

5.) How did they measure of visual field stabilization?

Response to 5.): We measured visual field stabilization by consecutive visual field tests. We added the criteria for visual field loss in the text (lines 81-84, section is underlined).

6.) The authors’ conclusion, namely trabecular aspiration has an effective medium-term pressure- lowering and medication-saving effect especially in patients without previous trabeculectomy is not supported. In the phaco combined trabecular aspiration group without trabeculectomy the lowering effect could also be explained with the deepening of anterior chamber after the lens removal. Only if a purely post-phaco pseudoexfoliative glaucomatous eye does not experience a significant reduction in intraocular pressure could it be said that trabecular aspiration has an effect. Trabecular aspiration might be a good adjuvant treatment possibility

Response to 6.): We agree with the Reviewer. In the conclusion of the abstract, we now changed “trabecular aspiration” to “Phaco-TA”. In the conclusion of the main text, we used already the term “Phaco-TA”.

Reviewer 3 Report

Comments and Suggestions for Authors

The authors retrospective analysised whether trabecular aspiration could lower IOP and reduce the need of medications in patients with PEG. They found that TA has an effective medium-term pressure-lowering and medication-saving effect especially in patients without previous trabeculectomy. In trabeculectomized eyes, the effect of TA is limited, but still could delay more invasive surgical intervention in some patients. Overall, the results are interesting. This study may provide new sights of TA for PEG patients.

The only problem is the medical records were form 2006 to 2012. Is there any new data could be included in?

Author Response

The authors retrospective analyzed whether trabecular aspiration could lower IOP and reduce the need of medications in patients with PEG. They found that TA has an effective medium-term pressure-lowering and medication-saving effect especially in patients without previous trabeculectomy. In trabeculectomized eyes, the effect of TA is limited, but still could delay more invasive surgical intervention in some patients. Overall, the results are interesting. This study may provide new sights of TA for PEG patients.

We thank very much the reviewer for these comments.

1.) The only problem is the medical records were from 2006 to 2012. Is there any new data could be included in?

Response to 1.): Thank you for your question. After 2012, various other microinvasive surgical procedures affecting aqueous humor outflow, such as miniaturized versions of trabeculectomy, ab interno canaloplasty, and various stents, have been introduced globally, and in our clinic, to treat PEG. This heterogenic group of “new” techniques was used either alone or in conjunction with cataract surgery to treat PEG. Therefore, trabecular aspiration was conducted infrequently or in a modified fashion in our clinic after 2012. Since most of these new devices turned out to be of limited effectiveness especially after previous trabeculectomy, we decided to analyze patients, who have been treated with an “older” minimally-invasive technique like trabecular aspiration to test if this technique is useful in patients with prior trabeculectomy. Although trabecular aspiration has been introduced in 1994, no data have been published so far on patients with prior trabeculectomy. We added a passage on this issue (lines 86-89, underlined).